# Learning Contextual Perturbation Budgets for Training Robust Neural Networks

**Jing Xu[1], Zhouxing Shi[2], Huan Zhang[2], Jinfeng Yi[3], Cho-Jui Hsieh[2], Liwei Wang[1]**
[1]Peking University  [2]UCLA  [3]JD AI Research
1700012451@pku.edu.cn; zshi@cs.ucla.edu; huan@huan-zhang.com;
yijinfeng@jd.com; chohsieh@cs.ucla.edu; wanglw@pku.edu.cn

## Abstract

Existing methods for training robust neural networks generally aim to make models uniformly robust on all input dimensions. However, different input dimensions are not uniformly important to the prediction. In this paper, we propose a novel framework to train certifiably robust models and learn non-uniform perturbation budgets on different input dimensions, in contrast to using the popular $\ell_\infty$ threat model. We incorporate a perturbation budget generator into the existing certified defense framework, and perform certified training with generated perturbation budgets. In comparison to the radius of $\ell_\infty$ ball in previous works, the robustness intensity is measured by robustness volume which is the multiplication of perturbation budgets on all input dimensions. We evaluate our method on MNIST and CIFAR-10 datasets and show that we can achieve lower clean and certified errors on relatively larger robustness volumes, compared to methods using uniform perturbation budgets. Further with two synthetic datasets constructed from MNIST and CIFAR-10, we also demonstrate that the perturbation budget generator can produce semantically-meaningful budgets, which implies that the generator can capture contextual information and the sensitivity of different features in input images.

## 1 Introduction

It has been demonstrated that deep neural networks, although achieving impressive performance on various tasks, are vulnerable to adverarial perturbations (Szegedy et al., 2013). Models with high accuracy on clean and unperturbed data can be fooled to have extremely poor performance when input data are adversarially perturbed. The existence of adversarial perturbations causes concerns in safety-critical applications such as self-driving cars, face recognition and medical diagnosis.

A number of methods have been proposed for training robust neural networks that can resist to adversarial perturbations to some extent. Among them, adversarial training (Goodfellow et al., 2015; Madry et al., 2018) and certified defenses (Wong et al., 2018; Gowal et al., 2018; Zhang et al., 2020) are of the most reliable ones so far, and most of them are trying to make the network robust to any perturbation within an $\ell_p$ norm ball. Taking the commonly used $\ell_\infty$-ball defense as an example, robust training methods aim to make model robust to $\epsilon$ perturbation on any pixel, which means the model is *uniformly robust* on all the input dimensions. But is this a valid assumption we should make?

As we know, human perception is non-uniform (humans focus on important features even though these features can be sensitive to small noise) and context dependent (what part of image is important heavily depends on what is on the image). We expect a robust model to be close to human perception, rather than learn to defend against a particular fixed threat model, e.g., the traditional $\ell_\infty$-norm one. Intuitively, we expect a good model to be more sensitive to important features and less sensitive to unimportant features, and the importance of features should be context-dependent. Taking the MNIST hand-written digit classification problem as an example, the digit 9 can be transformed to 4 simply by modifying just a few pixels on its head, so those pixels should be considered more important, and enforcing them to be robust to a large perturbation may not be correct. On the other hand, the pixels on the frame of such an input image can be safely modified without changing the ground-truth label of the image. Therefore, a uniform $\epsilon$ budget in robust training can greatly hamper the performance of

neural networks on certain tasks, and will force network to ignore some important features that are important for classification. Robustness certification with non-uniform perturbation budgets has been discussed in a prior work (Liu et al., 2019), but *training* robust models and *learning* context-dependent perturbation budgets has not been addressed in prior works, which is more challenging and important for obtaining robust models. A detailed discussion on our difference with Liu et al. (2019) is in Sec. 2.2.

In this paper, we propose the first method that can learn context-dependent non-uniform perturbation budgets in certified robust training, based on prior certified defense algorithms on $\ell_p$-norm threat models (Zhang et al., 2020; Xu et al., 2020). To learn a context-dependent budget without introducing too many learnable parameters, we introduce a perturbation budget generator with an auxiliary neural network, to generate the context-dependent budgets based on the input image. We also impose constraints on the generator to make generated budgets satisfy target robustness volumes and ranges of budgets, where robustness volume is defined as the multiplication of budgets on all input dimensions. We then train the classifier with a linear-relaxation-based certified defense algorithm, auto_LiRPA (Xu et al., 2020) generalized from CROWN-IBP (Zhang et al., 2020), to minimize the verified error under given budget constraints. The gradients of the loss function can be back-propagated to the perturbation budgets, allowing training the classification network and budget generator jointly in robust training.

Our contribution can be summarized below:

- We propose a novel algorithm to train robust networks with contextual perturbation budgets rather than uniform ones. We show that it can be incorporated into certified defense methods with linear relaxation-based robustness verification.

- We demonstrate that our method can effectively train both the classifier and the perturbation generator jointly, and we able to train models on relatively larger robustness volumes and outperform those trained with uniform budgets.

- We also show that the learned perturbation budgets are semantically meaningful and align well with the importance of different pixels in the input image. We further confirm this with two synthetic tasks and datasets constructed from MNIST and CIFAR-10 respectively.

## 2 BACKGROUND AND RELATED WORK

### 2.1 TRAINING ROBUST NEURAL NETWORKS

Since the discovery of adversarial examples (Szegedy et al. (2013), Biggio et al. (2013)), a great number of works has been devoted to improving the robustness of neural networks from both attack and defense perspectives (Moosavi-Dezfooli et al., 2016; Carlini & Wagner, 2017; Papernot et al., 2016; Moosavi-Dezfooli et al., 2017; Gowal et al., 2019). On a $K$-way classification task, training an adversarially robust neural network $f_w$ with weight $w$ can generally be formulated as solving the following min-max optimization problem:

$$\min_{w} \mathbb{E}_{(\mathbf{x},y)\sim\mathcal{D}} \max_{\delta\in\Delta} L(f_w(\mathbf{x}+\delta), y), \tag{1}$$

where $\mathcal{D}$ is the data distribution, and $\Delta$ is a threat model defining the space of the perturbations, and $L$ is a loss function.

Adversarial training (Goodfellow et al., 2015; Madry et al., 2018) applies adversarial attacks to solve the inner maximization problem and train the neural network on generated adversarial examples, with efficiency advanced in some recent works (Shafahi et al., 2019; Wong et al., 2020). However, robustness improvements from adversarial training do not have provable guarantees.

Some other recent works seek to train networks that have provable robustness, namely certified defense methods. Such methods solves the inner maximization by computing certified upper bounds that provably hold true for any perturbation within the threat model, including abstract interpretation (Singh et al., 2018), interval bound propagation (Gowal et al., 2018; Mirman et al., 2018), randomized smoothing (Cohen et al., 2019; Salman et al., 2019; Zhai et al., 2020), and linear-relaxation-based methods (Wong & Kolter, 2018; Mirman et al., 2018; Wong et al., 2018; Zhang et al., 2020; Xu et al., 2020). However, nearly all existing certified defense methods treat all input features equally in the

threat model, such as an $\ell_p$-ball threat models, $\Delta = \{\delta : \| \delta \|_p \le \epsilon\}$, especially the $\ell_\infty$ threat model commonly used in many previous works. But different input pixels are not uniformly important to the prediction, and thus we propose to learn contextual perturbation budgets for different pixels.

## 2.2 Robustness Certification with Non-Uniform Budgets

There has been a recent work Liu et al. (2019) studying robustness certification with non-uniform budgets. Their work aims to maximize the robustness volume while ensuring the network prediction is certifiably correct, and this optimization problem is solved using augmented Lagrangian method. To highlight, their method has several limitations: 1) The perturbation budgets are obtained by solving a constrained optimization problem for each input example, which is very time-consuming. This is not only too inefficient for training as it can bring a large overhead in each training step, but also incapable for learning perturbation budgets *jointly* with the classifier. In contrast, we have a significantly different scheme – we aim to maximize the accuracy under given target robustness volumes. And our perturbation budgets is obtained with a lightweight neural-network-based generator, and it can be jointly trained with the classifier in an end-to-end way. 2) Their work only focuses on certifying trained models due to the inherent limitation of the method, while we address of the problem of training robust neural networks and learning contextual perturbation budgets simultaneously. Consequently, we are able to empirically demonstrate that we can effectively train robust models with much larger robustness volumes and achieve lower errors under given target robustness volumes. 3) We further have experiments with synthetic tasks on MNIST and CIFAR-10 in Sec. 4.2 and Sec. 4.3 respectively to demonstrate that the learned perturbation budgets are semantically meaningful and can capture contextual information.

## 3 Proposed Method

### 3.1 Problem Setting

**Variable threat model** Unlike many previous works that use an $\ell_\infty$ threat model with a uniform $\epsilon$ on all input dimensions, we allow different pixels to have different perturbation budgets, but they need to meet some constraints as we will define below. For an $n$-dimensional input $\mathbf{x}$, when the perturbation budget of pixel $\mathbf{x}_i$ is $\epsilon_i$, the threat model is

$$\Delta(\epsilon_1, \epsilon_2, \cdots, \epsilon_n) = \{\delta : |\delta_i| \le \epsilon_i, 1 \le i \le n\}.$$

And thereby the $\ell_\infty$ threat model is a special case with $\epsilon_i = \epsilon_0 (\forall 1 \le i \le n)$. We define the *robustness volume* of a threat model $\Delta$ as the multiplication of all $\epsilon_i (1 \le i \le n)$:

$$V(\Delta) = \prod_{i=1}^{n} \epsilon_i, \quad \text{i.e., } \log V(\Delta) = \sum_{i=1}^{n} \log \epsilon_i. \tag{2}$$

In principal, we have a *target robustness volume*, $V_0 = \epsilon_0^n$, and $f_w$ is considered to be provably robust under this target robustness volume on instance $(\mathbf{x}, y)$ if and only if:

$$\exists \Delta \in \mathbb{D}, \forall k \ne y, \min_{\delta \in \Delta}([f_w(\mathbf{x} + \delta)]_y - [f_w(\mathbf{x} + \delta)]_k) > 0. \tag{3}$$

This means that the predicted score of the ground-truth class $y$ is certifiably larger than any other class $k \ne y$ under some threat model $\Delta$. And instead of using a fixed $\Delta$, in our framework $\Delta$ can be taken from a threat model space $\mathbb{D}$, which consists of threat models $\Delta(\epsilon_1, \epsilon_2, \cdots, \epsilon_n)$ satisfying the following two constraints:

$$\text{Volume constraint:} \qquad \sum_i \log \epsilon_i = n \log \epsilon_0, \tag{4}$$

$$\text{Range constraint:} \qquad l \le \epsilon_i \le u, \, l = \underline{\alpha} \epsilon_0, \, u = \min(\overline{\alpha} \epsilon_0, 1). \tag{5}$$

The volume constraint states that the robustness volume of $\Delta$ is equal to the target robustness volume $\epsilon_0^n$ (written in log domain above). For the range constraint, there are relative factors $\underline{\alpha}$ and $\overline{\alpha}$ controlling the perturbation budget range of each pixel, namely $[l, u]$. This constraint can be set to guarantee a minimum robustness and also prevent the model from over-invariant on each pixel.

**Perturbation budget generation**  A classifier can be accompanied by a perturbation budget generator $\epsilon(\mathbf{x})$ which tries to find perturbation budgets $\epsilon_1, \epsilon_2, \cdots, \epsilon_n$ and the corresponding threat model $\Delta(\mathbf{x})$ that can minimize the verified loss of the classifier while satisfying constraints (4) and (5), so that (3) is more likely to hold true with $\Delta$ generated by $\epsilon(\mathbf{x})$. We will state the optimization problem in the next paragraph.

**Robust classification**  Accompanied by a perturbation budget generator, we aim to learn a robust classifier $f_w$ with the following min-max optimization problem:

$$\min_{w} \mathbb{E}_{(\mathbf{x},y)\sim\mathcal{D}} \max_{\delta\in\Delta(\mathbf{x})} L(f_w(\mathbf{x}+\delta), y). \tag{6}$$

Note that a key difference between this problem and the traditional one in (1) is that now the threat model $\Delta(\mathbf{x})$ is variable and dependent on the input $\mathbf{x}$. This $\Delta(\mathbf{x})$ is generated by $\epsilon(\mathbf{x})$, under the given volume and range constraints. We evaluate the robustness of a classifier $f_w$ by computing an average verified error on all the test instances, where the verified correctness on each instance is evaluated similarly as (3), where $\Delta$ is taken as the generated $\Delta(\mathbf{x})$.

### 3.2  ALGORITHM FRAMEWORK

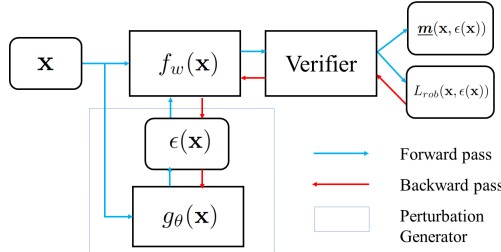

Figure 1: A flowchart of training a certifiably robust network with a perturbation budget generator. The dashed rectangle consists of the perturbation generator, which is based on an auxiliary neural network $g_\theta(\mathbf{x})$. The blue arrows and red arrows represent the forward pass and the back-propagation pass for gradient computation respectively.

Figure 1 illustrates our algorithm framework. Given an input $\mathbf{x}$, the perturbation budget generator outputs perturbation budgets $\epsilon(\mathbf{x})$ based on an auxiliary network $g_\theta(\mathbf{x})$ parameterized by $\theta$. Then a robustness verifier takes the unperturbed input $\mathbf{x}$ and perturbation budgets $\epsilon(\mathbf{x})$ as input and computes a robust loss function $L_{rob}(\mathbf{x}, \epsilon(\mathbf{x}))$. During training, this loss function is back propagated to compute its gradients w.r.t. the parameters of the classifier $f_w$ and also perturbation budgets $\epsilon(\mathbf{x})$, and the gradients w.r.t. $\epsilon(\mathbf{x})$ are further propagated the parameters of $g_\theta(\mathbf{x})$. $f_w$ and $g_\theta$ are jointly trained to minimize the robust loss. During test, the verifier also outputs a verified error for evaluation by computing the lower bound of the margin, $\underline{\mathbf{m}}(\mathbf{x}, \epsilon(\mathbf{x}))$ (see Sec. 3.4).

### 3.3  PERTURBATION BUDGET GENERATOR

For the perturbation budget generator $\epsilon(\mathbf{x})$, we generate perturbation budgets in two steps. In the first step, we generate initial perturbation budgets that are only required to meet the range constraint. And in the second step, we refine the perturbation budgets to meet the volume constraint. These two steps are detailed below.

**Initial perturbation budgets**  We first use a neural network $g_\theta(\mathbf{x}) \in [0, 1]^n$ to generate an *initial* budget distribution. Specifically, we assign $[\tilde{\epsilon}(\mathbf{x})]_i = \exp(\log l + [g_\theta(\mathbf{x})]_i(\log u - \log l))$ as the initial budget of the $i$-th pixel, and thereby $[\tilde{\epsilon}(\mathbf{x})]_i \in [l, u]$. In our work, we employ a two-layer convolutional model with a sigmoid activation in the last as $g_\theta(\mathbf{x})$.

**Refining the perturbation budgets**  We then refine the perturbation budgets to make them meet the volume constraint. We compute the log volume of these budgets, $\log V = n \log l + (\log u -$

$\log l) \sum_i [\tilde{\epsilon}(\mathbf{x})]_i$, and compare it with the target log robustness volume $\log V_0 = n \log \epsilon_0$. If $V < V_0$, we need to allocate $(V_0 - V)$ to the log perturbation budgets of different pixels, while still ensuring that each budget is no larger than $u$. This allocation is done proportionally according to the distance between the current log perturbation budget $\log[\tilde{\epsilon}(\mathbf{x})]_i$ and the upper limit $\log u$ of each pixel, for each pixel $i$, and then budget $\log[\epsilon(\mathbf{x})]_i$ is obtained with:

$$\log[\epsilon(\mathbf{x})]_i = \log[\tilde{\epsilon}(\mathbf{x})]_i + (n \log \epsilon_0 - V)\frac{\log u - \log[\tilde{\epsilon}(\mathbf{x})]_i}{n \log u - V} \qquad \text{for } V < V_0 = n \log \epsilon_0.$$

If $V > n \log \epsilon_0$, we remove redundant perturbation budgets. Similar to previous case, this refinement is done proportionally according to the distance between $\log[\epsilon(\mathbf{x})]_i$ and the lower limit $\log l$:

$$\log[\epsilon(\mathbf{x})]_i = \log[\tilde{\epsilon}(\mathbf{x})]_i - (V - n \log \epsilon_0)\frac{\log[\tilde{\epsilon}(\mathbf{x})]_i - \log l}{V - n \log l} \qquad \text{for } V > V_0 = n \log \epsilon_0.$$

### 3.4 CERTIFIED TRAINING WITH NON-UNIFORM PERTURBATION BUDGETS

For the robustness verifier, we adopt auto_LiRPA (Xu et al., 2020). It is based on linear relaxations and generalized from CROWN-IBP (Zhang et al., 2020), and there is also a loss fusion technique for more efficient robust training. In CROWN-IBP and also the default setting in auto_LiRPA, there are generally two passes for each input batch, excluding the gradient back-propagation pass. In the first pass, Interval Bound Propagation (IBP) (Mirman et al., 2018; Gowal et al., 2018) is used to compute the output bounds of intermediate layers, which is required by linear relaxations. Given an input $\mathbf{x}$ and a uniform perturbation budget $\epsilon$, the input of the IBP pass is the lower and upper bounds of the input image, $[Clip(\mathbf{x} - \epsilon), Clip(\mathbf{x} + \epsilon)]$, where $Clip(\cdot)$ stands for clipping the image bounds into domain $[0, 1]^n$. In the second pass, the CROWN algorithm (Zhang et al., 2018) is applied on the output layer to compute its linear bounds w.r.t. perturbed input $(\mathbf{x} + \delta)$, utilizing intermediate bounds by IBP for linearly relaxing activation functions. Let $h(\mathbf{x} + \delta)$ represent the output layer, then its linear bounds w.r.t. $(\mathbf{x} + \delta)$ and concrete (final) bounds without $(\mathbf{x} + \delta)$ are (Zhang et al., 2018):

$$\underline{\mathbf{A}}\mathbf{x} + \underline{\mathbf{b}} - \|\mathbf{A}\|_1\, \epsilon \leq \underline{\mathbf{A}}(\mathbf{x} + \delta) + \underline{\mathbf{b}} \leq h(\mathbf{x} + \delta) \leq \overline{\mathbf{A}}(\mathbf{x} + \delta) + \overline{\mathbf{b}} \leq \overline{\mathbf{A}}\mathbf{x} + \overline{\mathbf{b}} + \|\mathbf{A}\|_1\, \epsilon, \quad (7)$$

where $\underline{\mathbf{A}}, \overline{\mathbf{A}}, \underline{\mathbf{b}}, \overline{\mathbf{b}}$ are parameters of linear bounds obtained by CROWN. We refer readers to Xu et al. (2020) for their detailed algorithm.

When non-uniform perturbation budgets are used, $\epsilon(\mathbf{x})$ from the perturbation generator is used rather than scalar $\epsilon$. This causes differences mainly on the input lower and upper bounds for the IBP pass and the concretization of linear bounds in (7).

In inference, we compute the lower bound of the minimum margin between the ground truth label $y$ and any other label $i \neq y$:

$$\underline{\mathbf{m}}(\mathbf{x}, \epsilon(\mathbf{x})) = \min_{\delta \in \Delta(\mathbf{x})} \mathbf{m}(\mathbf{x} + \delta), \text{where } \mathbf{m}(\mathbf{x} + \delta) = \min_{i \neq y}([f_w(\mathbf{x} + \delta)]_y - [f_w(\mathbf{x} + \delta)]_i), \quad (8)$$

by taking the margin functions as the output layer in robustness verification. $\underline{\mathbf{m}}(\mathbf{x}, \epsilon(\mathbf{x}))$ is the lower bound of the margin for all possible perturbation $\delta \in \Delta$. The classifier is verifiably robust on this instance and the threat model $\Delta(\mathbf{x})$ from the generated perturbation budgets, if $\underline{\mathbf{m}}(\mathbf{x}, \epsilon(\mathbf{x})) > 0$, as defined in (3).

During training, we use loss fusion (Xu et al., 2020), to compute and optimize the robust loss $L_{rob}(\mathbf{x}, \epsilon(\mathbf{x}))$:

$$L_{rob}(\mathbf{x}, \epsilon(\boldsymbol{x})) \geq \max_{\delta \in \Delta(\mathbf{x})} L(f_w(\mathbf{x} + \delta), y), \quad (9)$$

where the robust loss is an upper bound of the right-hand-side, obtained from the verifier. The auto_LiRPA verifier is differentiable and thus we are able to back-propagate the gradients of the loss function w.r.t. the parameters of the both classifier and perturbation budget generator for training. In contrast, this joint training appears to be more difficult for adversarial training methods such as Projected Gradient Descent (PGD) attack (Madry et al., 2018). The PGD loss is technically differentiable to the perturbation budgets. However, take $N$-step PGD training as an example, each step we add a noise roughly $\frac{\epsilon}{N}$, so the $\epsilon$ is used by every step of PGD. Eventually, to get the correct gradient w.r.t $\epsilon$, we will need to backprop through the $N$ PGD update steps, which makes the gradient complicated and hard to calculate.

# 4 Experiments

We conduct experiments mainly on MNIST (LeCun et al., 2010) and CIFAR-10 (Krizhevsky et al., 2009) to demonstrate the effectiveness of our proposed method. We show that our method can effectively train certifiably robust models with learned contextual perturbation budgets, and the models achieve lower verified errors compared to those using uniform budgets. We also constructed two synthetic datasets to demonstrate that the learned perturbation budgets are semantically meaningful. We report implementation details in Appendix A.

## 4.1 Experiments on MNIST and CIFAR-10

For MNIST, we consider robustness volumes with $\epsilon_0 = 0.4, 0.6, 0.8$ respectively and set $\underline{\alpha}$ to 0.5 and $\overline{\alpha} = 2.0$, and for CIFAR-10, we consider $\epsilon_0 = 8/255, 32/255$ respectively and set $\underline{\alpha}$ to 0.125 and $\overline{\alpha}$ to 2.0. We reproduce baseline models with uniform budgets following the best setting in CROWN-IBP (Zhang et al., 2020), and in particular, the models with uniform perturbation budgets $\epsilon_0 = 0.4$ on MNIST and $\epsilon_0 = 8/255$ on CIFAR-10 are directly downloaded from their released models[1]. We evaluate the baseline models and our models with learned perturbation budgets mainly using clean error and verified error with the corresponding type of perturbation budgets used for training, i.e., baseline models are verified with uniform budgets and our models are verified with learned budgets. In addition, we also report verified error computed with Liu et al. (2019)[2], where the model is considered as verified on an example if and only if the volume of perturbation budgets by Liu et al. (2019) is at least $\epsilon_0^n$.

Table 1: Results of using uniform perturbation budgets (Zhang et al., 2020) and our learned ones respectively on MNIST for certified defense with various target robustness volumes. "-" denotes that training with uniform perturbation budgets $\epsilon_0 = 0.6$ or $\epsilon_0 = 0.8$ totally fails, and thus we also evaluate models trained with $\epsilon_0 = 0.4$ on test volumes $\epsilon_0 = 0.6$ and $\epsilon_0 = 0.8$ respectively. For verified errors in evaluation, we provide results evaluated with the method by Liu et al. (2019), on uniform budgets, and on our learned budgets respectively.

| Target Volume | Training budgets | Clean error (%) | Verified error (%) | | |
| --- | --- | --- | --- | --- | --- |
| | | | By Liu et al. (2019) | Uniform | Learned |
| $\epsilon_0 = 0.4$ | Uniform $\epsilon_0 = 0.4$ | 2.17 | **5.88** | 12.06 | 100.0 |
| | Learned $\epsilon_0 = 0.4$ (ours) | 1.41 | 7.97 | 58.10 | 7.81 |
| $\epsilon_0 = 0.6$ | Uniform $\epsilon_0 = 0.4$ | 2.17 | 100.0 | 100.0 | 100.0 |
| | Uniform $\epsilon_0 = 0.6$ | - | - | - | - |
| | Learned $\epsilon_0 = 0.6$ (ours) | 2.95 | 14.13 | 100.0 | **13.93** |
| $\epsilon_0 = 0.8$ | Uniform $\epsilon_0 = 0.4$ | 2.17 | 100.0 | 100.0 | 100.0 |
| | Uniform $\epsilon_0 = 0.8$ | - | - | - | - |
| | Learned $\epsilon_0 = 0.8$ (ours) | 12.15 | **25.77** | 100.0 | 26.37 |

Table 2: Results on CIFAR-10, in a similar format with Table 1.

| Target Volume | Training budgets | Clean error (%) | Verified error (%) | | |
| --- | --- | --- | --- | --- | --- |
| | | | By Liu et al. (2019) | Uniform | Learned |
| $\epsilon_0 = 8/255$ | Uniform $\epsilon_0 = 8/255$ | 54.02 | 64.11 | 67.11 | 67.75 |
| | Learned $\epsilon_0 = 8/255$ (ours) | 49.34 | 65.99 | 86.42 | **60.72** |
| $\epsilon_0 = 32/255$ | Uniform $\epsilon_0 = 8/255$ | 54.02 | 83.83 | 99.57 | 99.96 |
| | Uniform $\epsilon_0 = 32/255$ | 80.63 | 84.73 | 85.03 | 85.11 |
| | Learned $\epsilon_0 = 32/255$ (ours) | 63.34 | 82.0 | 98.56 | **69.31** |

Results are shown in Table 1 for MNIST and Table 2 for CIFAR-10. On MNIST, on the smaller $\epsilon_0 = 0.4$ target volume, our model has lower clean error and lower verified error on learned budgets, compared to the model trained and evaluated with uniform budgets, though the verified error of our model is higher than the one trained with uniform budgets when evaluating with Liu et al. (2019). When using larger target volumes, we find that training with uniform budgets totally fail on $\epsilon_0 = 0.6$ and $\epsilon_0 = 0.8$, because we are able to totally break the classification by perturbing every pixel to

---

[1]`https://github.com/huanzhang12/CROWN-IBP`
[2]`https://github.com/liuchen11/CertifyNonuniformBounds`

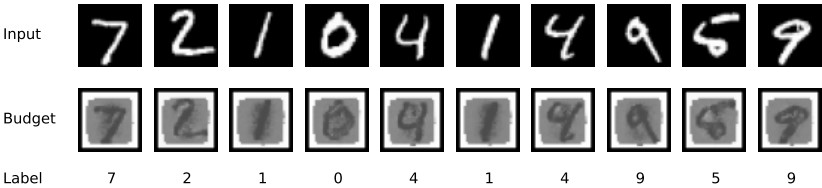

Figure 2: Visualization of learned perturbation budget distributions on MNIST with $\epsilon_0 = 0.6$. The *budget* images have pixels ranging from 0 to 1 representing a per-pixel, per-example $\epsilon$. Lighter colors indicate larger perturbations and darker colors indicate smaller perturbations.

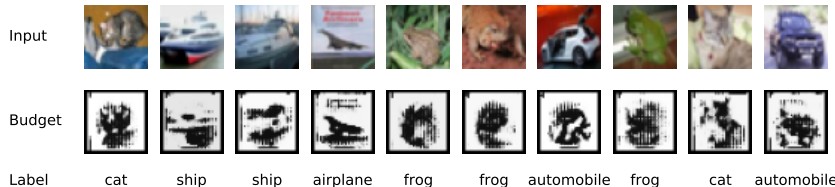

Figure 3: Visualization of learned perturbation budget distributions on CIFAR-10 with $\epsilon_0 = 16/255$. For better visualization, the visualized budgets are normalized by the upper limit of the perturbation budgets, 32/255, since the budgets on CIFAR-10 are much smaller than 1.

0.5 under uniform perturbation budgets $\epsilon = 0.6$ or $\epsilon = 0.8$, and the model trained with $\epsilon_0 = 0.4$ uniform budget is not able to generalize to $\epsilon_0 = 0.6$ and $\epsilon_0 = 0.8$ target robustness volumes in test. In contrast to using uniform budgets, with our learned budgets, we are able to handle $\epsilon_0 = 0.6$ and $\epsilon_0 = 0.8$, achieving reasonable verified errors. Moreover, on CIFAR-10, our models using learned budgets achieve lower verified errors than models with uniform budgets on both $\epsilon_0 = 8/255$ and $\epsilon_0 = 32/255$. The experiments demonstrates the effectiveness of our method for training more robust models under non-uniform perturbation budgets.

We note that our models have higher verified errors when evaluated on uniform budgets, but evaluation on uniform budgets is not the major goal of this paper, nor it is the ultimate goal of robust machine learning, as we have argued that non-uniform budgets are more practical. Nevertheless, in case that one expects to maintain a performance on uniform budgets while training with non-uniform learned budgets, this can be achieved by setting $\underline{\alpha}$ which controls the minimum budget of each pixel, as we demonstrate in Appendix C.

We also visualize our learned perturbation budget distributions in Figure 2 and Figure 3 for MNIST and CIFAR-10 respectively. Our perturbation budget generator is able to identify important and sensitive features in the input image and assign relatively smaller budgets to the corresponding pixels, while insensitive pixels are assigned with relatively larger perturbation budgets. Note that the perturbation budgets also highlight sensitive pixels that are missing from input images. Therefore, learned perturbations contain important features from both the ground truth class and the classes that are close to it. For example, as visualized in Figure 2, perturbation budgets of digit 4 combine features from both 4 and 9, and digit 5 looks like 5 overlapped with 8.

## 4.2 Experiments on Watermarked MNIST

To demonstrate another benefit of using different perturbation budgets for different pixels, we artificially constructed a dataset from MNIST, namely the Watermarked MNIST. For each image in the original MNIST dataset, as shown in the first row of Figure 4, we add a small watermark to the left or right margin of the image, which decides the gold label jointly with the main digit in the image. The watermark is darker than the main digit, and thus we expect to see the perturbation generator assign smaller perturbation budgets to the watermark, to retain the information carried by the watermark, while training uniform budgets is likely to fail for breaking the watermark. Specifically, the watermark is a digit of zero or one, chosen randomly, from an image with the corresponding label in the original dataset. We scale down the image to $12 \times 12$, and we crop out 2 pixels on each border and retain the

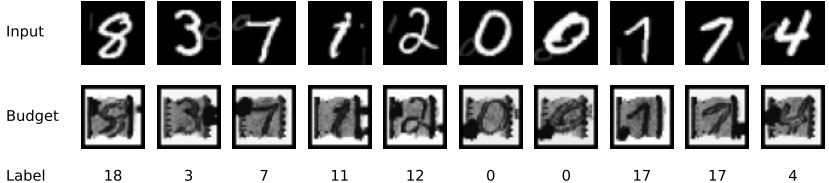

Figure 4: Images in the artificial Watermarked MNIST dataset and the visualization of our learned perturbation budget distributions. The watermarked pixels must use a smaller perturbation to be correctly recognized, and our learned perturbation correctly chooses a smaller perturbation budget for these pixels.

inner $8 \times 8$ part. The opacity of the watermark is reduced to 30%. We randomly put the watermark either on the left or right $28 \times 8$ border, and the vertical position span is randomly chosen from $[0, 7], [1, 8], \cdots, [20, 27]$. The gold label of the new image is the main digit itself, if the watermark is 0, otherwise the gold label is the main digit *plus* 10. In this way, we construct a 20-class digit classification problem.

We then perform certified defense on this synthetic dataset, with $\epsilon_0 = 0.4$. For our learned perturbation budget generator, we set $\underline{\alpha} = 0.125, \overline{\alpha} = 2.5$. We compare the results with the method using uniform perturbation budgets, as shown in Table 3. When uniform perturbation budgets are used, the clean error and the verified error are very large (higher than 50%). This is because the values of the pixels with the watermark is around 30% of those in an original digit, and thus when a perturbation budget of $\epsilon \geq 0.3$ is uniformly added to all the pixels, it is enough to perturb the image and make the watermark disappear, and thus the model cannot achieve low errors. In contrast, as shown in Figure 4, by learning contextual perturbation budgets, the budget generator tends to assign relatively small budgets to pixels with the watermark. Pixels with the main digit tend to have budgets that are somewhat larger, and there are empty pixels which receive even larger budgets. With such perturbation budgets, we achieve much lower verified error (19.04 v.s. 59.29 on $\epsilon_0 = 0.4$) and clean error (3.48 v.s. 51.75 on $\epsilon_0 = 0.4$) under the same robustness volume, as shown in Table 3. This experiment demonstrates the importance of learning different perturbation budgets for different pixels and the effectiveness of our proposed method.

Table 3: Results on the artificial Watermarked MNIST dataset. Using an uniform budget resulting in a $> 50\%$ error, as the classifier are not sensitive enough to these watermarked pixels.

| Target Volume | Uniform budgets (Zhang et al., 2020) | | Learned budgets (Ours) | |
|---|---|---|---|---|
| | Clean error (%) | Verified error (%) | Clean error (%) | Verified error (%) |
| $\epsilon_0 = 0.3$ | 50.90 | 55.54 | **1.99** | **11.00** |
| $\epsilon_0 = 0.4$ | 51.75 | 59.29 | **3.48** | **19.04** |

### 4.3 EXPERIMENTS ON DOUBLED CIFAR-10

To further demonstrate the ability of the perturbation generator in identifying important input features, we create a harder synthetic dataset, namely *Doubled CIFAR-10*, based on CIFAR-10. An original input image in CIFAR-10 has a size of $32 \times 32$. In this setting, each new input image consists of two original images on the left and the right respectively. Two images combined in this way yield a new one with a size of $32 \times 64$. We also add a $4 \times 64$ border to the top of the new image and make the border either red or green. If the border if red, the gold label is taken as the gold label of the left original image, otherwise it is the gold label of the right original image. The border serves as a hint for the model to tell which original $32 \times 32$ image it should look at for prediction. Thereby the size of each new image is $36 \times 64$. And with such a dataset, we expect to further study whether the perturbation budget generator can identify important features in the input image, since now one of the two component image in each new image does not affect the gold label, and thus the model is expected to assign large perturbation budgets on this component image.

We use $\epsilon_0 = 16/255$, $\underline{\alpha} = 0.0625$, $\overline{\alpha} = 1.5$. Results are shown in Table 4. Again, the errors of the model with learned perturbation budgets are much lower than those of the model with uniform

Table 4: Results on the synthetic Doubled CIFAR-10 dataset.

| Uniform budgets (Zhang et al., 2020) | | Learned budgets (Ours) | |
|---|---|---|---|
| Clean error (%) | Verified error (%) | Clean error (%) | Verified error (%) |
| 73.16 | 81.29 | **56.11** | **63.24** |

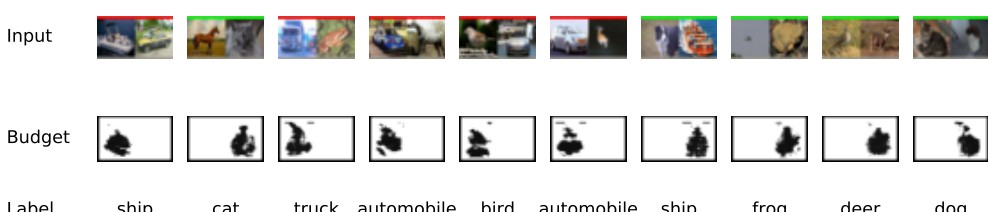

Figure 5: Images in the synthetic Doubled CIFAR-10 dataset and the visualization of our learned perturbation budget distributions. The visualized budgets are normalized by $1.5 \times 16/255$ similarly as Figure 3. The generated perturbations correctly correlate with border colors.

budgets. We also visualize the learned perturbation budgets in Figure 5. We observe that when the border is in red, large perturbation budgets are assigned to the right side, which is consistent with that the gold label is determined only by the left side in this case. This observation also holds true for the case when the border is in green. Moreover, we also observe that the perturbation budget distributions of the other side are semantically meaningful. This experiment further demonstrates that our proposed method is able to identify important features in the input and learn semantically meaningful distributions of perturbation budgets.

## 5 CONCLUSIONS

In this paper, we propose a novel approach to learn contextual perturbation budgets and successfully incorporate it into linear-relaxation-based certified training methods. We show its superiority over methods with uniform perturbation budgets via experiments on MNIST and CIFAR-10. We also demonstrate that the perturbation budget generator can capture semantic information and align well with features in the input image with synthetically constructed Watermarked MNIST and Doubled CIFAR10 datasets. Our proposed method can be used in settings where the perturbation budget of a dataset cannot be easily determined as in toy datasets, and can handle context dependent perturbations.

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

# A    IMPLEMENTATION DETAILS

We conduct experiments on classification models with similar structures as were used in prior works (Gowal et al., 2018; Zhang et al., 2020). Specifically, we adopt the `DM-small` and `DM-large` models used in CROWN-IBP (Zhang et al., 2020). They are both convolutional models with 2 and 5 convolutional layers respectively, and their details have been reported in Zhang et al. (2020). On CIFAR-10, we use `DM-large`, and on MNIST, we find that `DM-large` has a serious overfitting problem when training with learned perturbation budgets, and thus we use `DM-small` instead. Nevertheless, we still use `DM-large` to report the results of models with uniform budgets following the best setting in CROWN-IBP.

For the perturbation budget generator, it is a two-layer convolutional model with ReLU activations after each convolutional layer, and the two convolutional layers have 4 and 8 channels respectively and a kernel size of 3. After that, there is a linear layer mapping the hidden layer to each pixel in the image, with a Sigmoid activation for computing initial perturbation budgets.

For training with certified defense, we mainly follow settings in CROWN-IBP (Zhang et al., 2020), but we use the loss fusion technique proposed in auto_LiRPA for faster training (Xu et al., 2020). There is a scheduling on the robustness volume during training, similar as the scheduling of uniform perturbation radius in CROWN-IBP. On MNIST, we use a linear scheduler, where the robustness volume linearly increases from 0 to the final target robustness volume in 60 epochs starting from the 2nd epoch. The model is trained for 100 epochs. On CIFAR-10, we use a smoothed scheduler with a length of 160 epochs starting from the 32nd epoch, which slowly morphs into a linear schedule (Gowal et al., 2018). The model is trained for 320 epochs.

# B    VISUALIZATION ON MEDMNIST DATASET

MedMNIST (Yang et al., 2020) is a recently released dataset consisting of 10 subsets of lightweight $28 \times 28$ medical images. In this section, we utilize the OrganMNIST(axial) subset consisting of CT images of body organs cropped from Abdominal CT images (Bilic et al., 2019; Xu et al., 2019) as an example, to further justify why non-uniform contextual perturbation budgets are needed in training robust models in a more practical scenario. In such medical images, classifiers need to identify and classify abnormal regions in body organs. In these complex medical images, some regions should be rather sensitive to adversarial perturbations, e.g., if regions right correspond to a specific abnormality is perturbed, the image label is very likely to change, while some other regions may be able to tolerate larger perturbations. This is a practical situation where it is supposed to especially useful to leverage non-uniform and contextual perturbation budgets for robust training. For an initial demonstration, we train a model with $\epsilon_0 = 16/255$ using learned contextual perturbation budgets and uniform budgets respectively. For the model with learned budgets, the clean error is 21.8 and the verified error is 50.6, while for the model with uniform budgets, the clean error is 28.0 and the verified error is 57.5. Using learned contextual perturbation budgets, we are able to obtain a model with lower clean and verified errors under the same target robustness volume. We also visualize several input images and our learned perturbation budgets in Figure 6. The learned perturbation budgets appear to be semantically meaningful and align with the input images.

# C    CONTROLLING PERFORMANCE ON UNIFORM PERTURBATION BUDGETS

In our experiment results shown in Table 1 and Table 2, we find that models trained with our learned budgets tend to have higher verified errors when the models are evaluated with uniform budgets. Although we have argued that evaluation on uniform budgets is not our major goal and it is less practical than learned non-uniform budgets, in this section, we demonstrate that it is still possible to maintain a performance on uniform budgets while training with learned budgets by controlling $\underline{\alpha}$. We train a model on MNIST with $\epsilon_0 = 0.5$ and $\underline{\alpha} = 0.6$, which means the minimum budget of each pixel is 0.3. As a result, verified error on non-uniform budgets is 10.46, while the verified error on uniform $\epsilon = 0.3$ budget is 9.96, which is close to the 9.40 verified error reported in CROWN-IBP (Zhang et al., 2020) on the same "DM-small" model trained with uniform budgets. And if $\epsilon_0 = 0.6, \underline{\alpha} = 0.5$, the verified error on uniform $\epsilon = 0.3$ is 11.22 which is also not far away from 9.40, while now our model has a robustness on a volume as large as $0.6^n$.

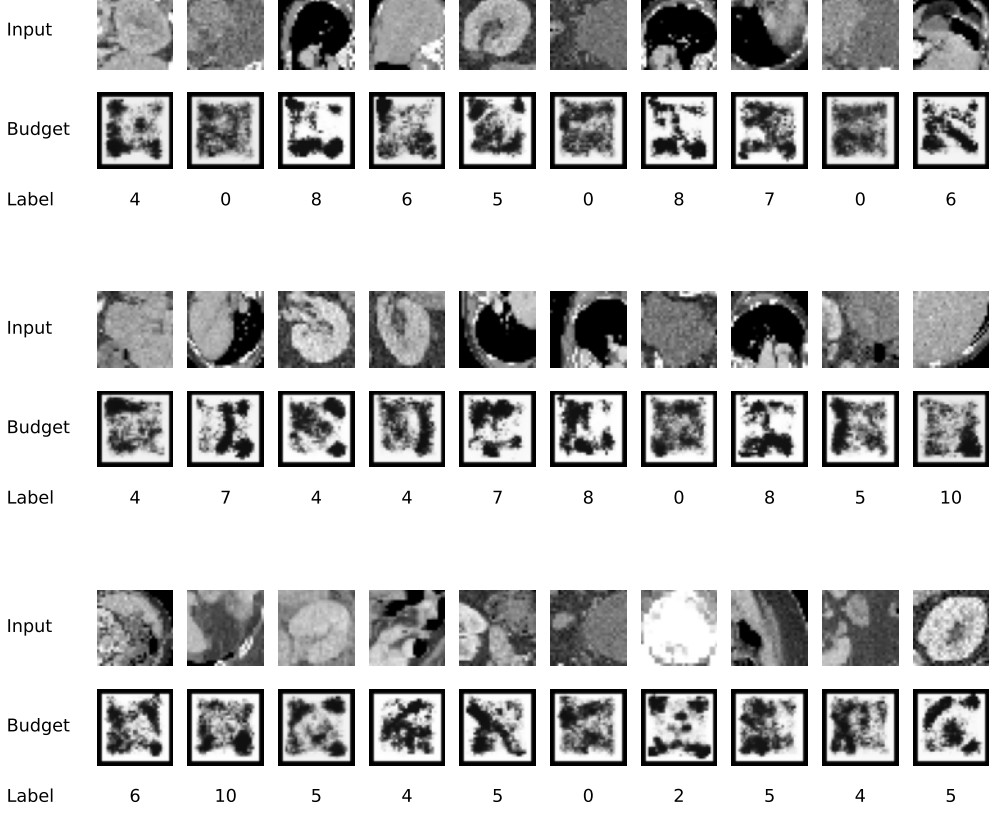

Figure 6: Visualization of input images and learned perturbation budgets on the OrganMNIST(axial) subset in the MedMNIST dataset.

