# OpenReview forum: "Learning Contextual Perturbation Budgets for Training Robust Neural Networks"
_ICLR.cc/2021/Conference — Reject_

### Official Review · AnonReviewer2 · 2020-10-22
**Review for "Learning Contextual Perturbation Budgets for Training Robust Neural Networks"**

**Rating:** 5
**Confidence:** 5

**Review:**

Update after the rebuttal:

I read response from the authors and other reviews. I have increased my score to 5 given that authors now performed some
comparison with Liu et al. However, I still believe that the threat model is not realistic and that attacker can not be bounded by the
budget that is produced by generator network here. While authors wrote quite detailed response, I did not find it convincing enough. As R5 points out similar issues, I would encourage authors to think more on how to tackle the problem better. Perhaps the threat model can limit the attacker to all possible perturbations under the certain volume budget, but that would be quite different from the idea in this paper.  Thus, I can not recommend acceptance at the current state.

============================================================================

-> Summary:

Authors propose to train provably robust models with respect to the threat model
that assumes non-uniform perturbation budget for the attacker. Under this threat model,
attacker is allowed to perturb some pixels more than the others, unlike for the standard
l_p threat model here each pixel has a maximum perturbation value. In this work, authors
propose perturbation generator which produces bounds for each pixel and certified defense
to train provably robust model with respect to the generated perturbation bounds. They show
that trained models have lower natural and verified error than models trained with uniform
bounds of equivalent volume.

-> Reasons for score:

I vote for rejecting this paper. The biggest issue I have is that authors do not compare (and do not cite)
with the work of Liu et al. [1] whose contributions I believe substantially overlap with contributions of
this submission. The other concern I have is that the idea of learning a perturbation budget which
the attacker should obey seems somewhat unrealistic.

-> Pros:

- I like the general idea of a threat model where maximum perturbations is different for each pixel.
- Paper is well written, easy to follow and manages to bring across the main points.
- The authors perform experiments on several datasets and present visualizations which help to understand
what the method has actually produced.

-> Cons:

- The biggest issue I have with this paper is omission of a prior work of Liu et al. [1] which
was the first paper to consider non-uniform perturbation bounds. Liu et al. consider same
volume constraint as in Equation (4) in this submission and propose a Lagrangian method to maximize
log of the volume. I think it is essential that authors compare their perturbation generator in section 3.2
with the Lagrangian method from Liu et al. and check which of the two methods can generate larger volume.
One contribution that this submission has and Liu et al. does not is that of certified defense as Liu et al.
focus only on certification of already trained networks (I am not sure if their method can be trivially extended
for training). However, given that authors simply use auto_lirpa library for certified defense, contributions there
are also limited.
- I am not sure that idea of generating perturbation budget is feasible in practice. What happens here
is that we generate the threat model under which attacker operates by ourselves and then we somehow expect
that attacker should obey this threat model. For example, let's say that proposed procedure results in
eps(x) = [0.3, 0.1, 0.1]. Why would attacker have to obey the fact that third pixel is perturbed by at most 0.1?
Essentially, the method here learns eps(x) such that attacker can't find adversarial example, but I don't understand
in what scenario would attacker be limited by the *particular* eps(x) that this method has produced?
I think it would be more sensible to have a defense which guarantees that model is robust under all threat models
which have volume below some constant V_0, but on the other hand this would be strictly more difficult than the
uniform baseline.
- Another thing I think is missing is some baseline for computing perturbation budget introduced in 3.3.
This relates to work of Liu et al. who proposed Lagrangian method to do this. Given that algorithm in 3.3
is perhaps the biggest contribution of this work, I would have expected there to be at least some baseline to compare with.

[1] Liu, Chen, Ryota Tomioka, and Volkan Cevher. "On certifying non-uniform bound against adversarial attacks." ICML 2019.


-> Questions:

- Can authors comment on the work of Liu et al. [1] and what they think are main contributions of their work
which are not already present in Liu et al.? Ideally, authors should also compare to their method (at least
certification part as they have no training).
- Can you explain what would be realistic scenario where attacker has to obey perturbation budget generated by this method?

-> Minor comments:

- typo: "generailized"
- In "Refining the perturbation budgets", third line, should it be [g_\theta(x)]_i under the summand?

---

> ### Author Response · Authors · 2020-11-22
> **Additional experiments and about minor comments (3/3)**
>
> ### Additional Experiments
> 1. Evaluating models using Liu et al, 2019’s method
>
> We have reproduced the algorithm proposed in Liu et al., 2019 with their open-source code, and the new results have been updated to **Table 1 and Table 2**. On MNIST, when models are evaluated by Liu et al., the model trained with $\epsilon=0.4$ uniform budget has a little lower verified error (5.88 v.s. 7.97) than the model with learned budgets. However, as the target volume is increased, on $\epsilon_0=0.6$ and $\epsilon_0=0.8$, models trained with uniform budgets totally fail with verified error 100.0%, while our models can still achieve reasonable verified errors (13.93 on $\epsilon_0=0.6$ and 25.77 on $\epsilon_0=0.8$). On CIFAR-10, our models can **achieve lower verified errors than models trained with uniform budgets**, even if the uniform budget models are evaluated with Liu et al.
>
> 2. MedMNIST
>
> To show the importance of using a non-uniform perturbation budget in a more realistic setting, we conduct additional experiments on the MedMNIST dataset. MedMNIST [3] is a recently developed medical imaging dataset. There are 10 sub-datasets and we adopt the OrganMNIST(axial) subset consisting of body organ CT images (please refer to Appendix B for more details). On target robustness volume $\epsilon_0=16/255$, we train a model with learned perturbation budgets and uniform budgets respectively. For the model with learned budgets, the clean error is 21.8 and the verified error is 50.6, while for the model with uniform budgets, the clean error is 28.0 and the verified error is 57.5. Using learned contextual perturbation budgets, we are able to obtain a model with lower clean and verified errors under the same target robustness volume. We visualize learned budgets in Figure 6 in Appendix B.
>
>
> ### About minor comments
>
> 1. We thank the reviewer for pointing out the typo.
> 2. About the second “minor comment”, we think ours is correct. $g_\theta(x)$ is generated by the generator network, but the generated value is relative to $l$ and $u$ range constraints and not absolute perturbation budgets. $g_\theta(x)$ is turned into perturbation budgets $\tilde{\epsilon}(x)$ as shown in the “Initial perturbation budgets” paragraph, and thus it is $\tilde{\epsilon}(x)$ in the summation you mentioned.
>
>
> ### Conclusion
> We hope our responses can address your questions and concerns about the paper. We would also be happy to answer any other questions you may have.
>
>
> References:
>
> [1] Tramèr F, Behrmann J, Carlini N, et al. “Fundamental tradeoffs between invariance and sensitivity to adversarial perturbations” ICML 2020
>
> [2]Liu, Chen, Ryota Tomioka, and Volkan Cevher. “On certifying non-uniform bound against adversarial attacks.” ICML 2019.

---

> ### Author Response · Authors · 2020-11-22
> **Justification on non-uniform perturbation budgets (2/3)**
>
> ### Why the attackers have to obey non-uniform perturbation budget
>
> The review is concerned that a real-world attacker would not obey the perturbation budgets generated by our method. However, real world adversaries do not have to restrict themselves to the popular $\ell_p$ threat model as well. The use of the  $\ell_p$ threat model is mostly for mathematical convenience and is sometimes problematic in real settings. In fact, the contextual perturbation budget in our method is a more realistic threat model for the following reasons:
>
> First, real world attackers indeed have to **follow the semantics** of the input in order to produce meaningful perturbations. The goal of our perturbation generator is to learn “what is the maximal allowable perturbation **that preserves ground-truth label given the current context**”? A successful attack has to achieve the following two goals: fooling the model and maintaining the ground truth label of the input. If the attacker simply disregards the semantics of the input image, the second condition can be violated, for the perturbed image will either be too messy to look like a real image, or the ground truth label of the perturbed image will be different from that of the original image (e.g., see the “invariance adversarial examples” in [1]), and this is not an adversarial attack anymore.
>
> Ideally, **an attacker needs to obey our learned perturbation budget, to avoid changing the ground-truth label**. Existing works use a simple $\ell_p$ norm to determine if the ground-truth model is changed or not. Because the $\ell_p$ threat model treats each pixel equally, while different features in the input image in fact have quite different importance depending on context, and [1] has demonstrated the cases that the ground-truth label can change within $\ell_p$ norm perturbation budget. Our non-uniform defense, in contrast, is semantically meaningful as more important pixels are assigned a smaller perturbation radius according to our contextual perturbation budgets. This is in fact a more realistic and useful threat model than $\ell_p$ norm.
>
> Ultimately, **the goal of robust machine learning is to learn a classifier close to human perception, rather than “robust” to certain $\ell_p$ norms**. Human perception is non-uniform (humans focus on important features even though these features can be sensitive to small noise) and context dependent (what part of image is important heavily depends on what’s on the image). Admittedly, the perturbation generation process in our paper still has large room to improve to eventually match human perception. However, we believe it is import to think beyond the $\ell_p$ robustness model for robust training, especially in our work we consider *context dependent* perturbation budget, which was never proposed by prior works. We hope our paper can help the community understand the limitation of $\ell_p$ norm and think beyond it.
>
> Additionally, another important motivation for our work is that real datasets have varied sensitivity on each feature, and our method enables robust training on these datasets effortlessly. In contrast, the $\ell_p$ threat model is problematic because it is hard to define an uniformly good perturbation budget, and such a budget can be feature dependent and contextual so not easily defined manually. This prevents the usage of robust training in practical settings. For example, in a medical imaging setup, the difference between health and unhealthy tissues may be subtle and can only be recognized by experts. Using an uniform $\ell_\infty$ perturbation for robust training is likely to destroy the subtle but important features, reducing clean accuracy. Training on a uniform radius can harm the clean accuracy of the model. Empirically, we have reported initial results on the MedMNIST dataset to support this point (see details below).

---

> ### Author Response · Authors · 2020-11-22
> **Reply to AnonReviewer2: Novelty and comparison to Liu's paper (1/3)**
>
> **(There are currently 3 posts in total for our author response to AnonReviewer2. This is the first one, and the other 2 are presented below this one.)**
>
> Dear AnonReviewer2,
>
> Thank you for your efforts in reviewing our paper and providing insightful suggestions. In our response, we provide a detailed comparison to the method of Liu et al., provide additional experiments and give justifications for why the attacker has to obey our contextual perturbation budgets.
>
> Here are our detailed responses to your questions and concerns:
>
> ### Novelty of our method and comparisons to Liu’s paper
>
> We apologize for not citing this insightful paper. We have updated our paper which cites the work of Liu et.al [2]. Despite both Liu's method and our method considering non-uniform perturbation bounds, we have made significantly different contributions. We summarize the differences and provide additional experiments as follows:
>
> 1. We are a robust training method:
>
> In Liu’s work, they propose to compute a largest possible certified volume for a **pretrained model**. They don’t update model parameters. In contrast, our method can **train the classifier jointly with the generator**, so that our model can learn semantically meaningful perturbation budgets.
>
> Additionally, Liu’s method cannot be extended to an efficient training method, as their method requires solving a constrained optimization problem via the Lagrangian method for each input example using a large number of iterations, which is too costly for training. We introduce a perturbation generator network with nearly no additional cost in training compared with existing uniform certified defense methods.
>
> 2. Different problem formulation:
>
> Liu’s method **maximizes the volume** of the perturbation budget for a fixed network under the constraint that the network prediction is correct within this perturbation region, while our method fixes the volume of the perturbation budget and jointly trains a perturbation generator with the classifier to **maximize the robust accuracy**, which matches the training objective of prior works for certified defense.
>
> 3. Certification approach and efficiency:
>
> As for certification in the inference stage, Liu’ method is still too inefficient as a large number of optimization iterations are required for every batch, while we only need a forward propagation with a small perturbation generation network, and use the efficient (CROWN-)IBP method to obtain certificates.
>
> 4. Semantic and contextual perturbation:
>
> Our visualization of the generated perturbation budget and experiments on Watermarked MNIST and Doubled CIFAR-10 (see section 4.2,4.3 for details) demonstrates that our perturbation budgets indeed learns contextual knowledge. This implies that training the classifier and the generator jointly enables the budget generator to capture the contextual information of input images, rather than just optimizing training objectives, while Liu’s work did not have such analysis.
>
> 5. Additional experiments:
>
> In Section 4, we conduct additional experiments to evaluate models with the method in Liu et al., as also shown in the “New experiments and comparisons” section in this response. Our method is capable of training robust models on larger robustness volumes and achieving lower verified errors.

---

### Official Review · AnonReviewer1 · 2020-10-27
**Review for "Learning Contextual Perturbation Budgets for Training Robust Neural Networks"**

**Rating:** 6
**Confidence:** 2

**Review:**

This paper proposes to change the perturbation budget for adversarial attacks to a non-uniform setting where differet input pixels have different perturbation budgets. To achieve this, an additional network is trained to learn the perturbation budget for each part of the input. The approach seems to perform better than a uniform perturbation budget and also learns semantically meaningful budgets for the input.

I am not an expert on this topic and will, therefore, keep my review quite short.

The idea that not all parts of the input should be treated equally makes sense and is well motivated.

Questions/remarks:
- What is the exact architecture of the network that learns the perturbation budged? Is it purely convolutional? Will it easily scale to larger inputs?
- It would be interesting to see the performance of your model on more complicated datasets, e.g. Tiny-ImageNet
- How much overhead (training time, model size, etc) does the training of the additional network for the perturbation budget introduce?
- For the visualization of learned pertubation budget: it would be interesting to also run some analysis/visualization of what parts of the input have the strongest effect on the final prediction to see if this correlated with your learned perturbation budged (i.e. if the parts of the input that have the strongest effect on the final classification also have the smallest perturbation budget)

---

> ### Author Response · Authors · 2020-11-22
> **Reply to comments by AnonReviewer1**
>
> Dear AnonReviewer1,
>
> Thank you for carefully reading our paper and we address your helpful comments below:
>
> 1. Network architecture of the generator:
>
> We use a simple two-layer convolutional neural network in the generator. It was previously included in Sec 4.1 in the initial version of our paper, and now we have moved it with other implementation details to Appendix A. The architecture is purely convolutional and can be easily scaled to large inputs by changing the input parameters.
>
> 2. Larger dataset:
>
> In our paper, we focus on certified adversarial defense with provable robustness, but unfortunately these methods can only scale to relatively small datasets. For example, for CIFAR-10, the state of the art method needs to train on 32 TPUs for a few hours [1], or on 4 GPUs for 1 day. This approach is scaled to larger dataset such as TinyImageNet in a very recent work [2]. Due to limited time in the discussion period, we were not able to conduct new experiments on TinyImageNet, but technique developed in [2] can be applied to us as well.
>
> 3. Overhead of our method.
>
> The size of the perturbation budgets generator, which is a simple two-layer convolutional neural network in our experiments as mentioned above, is small compared with the size of the classifier which has up to 5 convolutional layers. What’s more, our algorithm enables efficient joint training of the classifier and the generator, so our non-uniform perturbation budget generator will introduce very little additional costs compared with methods with uniform budgets. Our experiments also demonstrate that a simple convolutional neural network suffices to capture the contextual information in MNIST and CIFAR10 datasets.
>
> Empirically, we have evaluated the training time: on MNIST, it is 11.02s/epoch for the uniform one and 12.28s/epoch for the non-uniform one; on CIFAR-10, it is 14.62s/epoch for the uniform one and 16.72s/epoch for the non-uniform one. Experiments are done on an Nvidia GTX 1080Ti GPU. The measured overhead is only around 15%. For the model size, on MNIST, there are 4.9M parameters in the generator and 13.3M parameters in the classifier; on CIFAR-10, there are 8.4M parameters in the generator and 17.1M parameters in the classifier. The majority of parameters are in the last fully-connected layers.
>
> 4. Visualization:
>
> To demonstrate that our perturbation budget generator can indeed identify sensitive (important) features and choose a smaller perturbation budget for these pixels, we construct two artificial datasets, namely Watermarked MNIST and Doubled Cifar10, presented in Section 4.3 and 4.4. Our results show that the learned perturbation budget is highly correlated with the important features in the input image.
>
> References:
>
> [1] Huan Zhang, Hongge Chen, Chaowei Xiao, Bo Li, Duane Boning, and Cho-Jui Hsieh. Towards stable and efficient training of verifiably robust neural networks. In International Conference on Learning Representations, 2020.
>
> [2] Kaidi Xu, Zhouxing Shi, Huan Zhang, Yihan Wang, Kai-Wei Chang, Minlie Huang, Bhavya
> Kailkhura, Xue Lin, and Cho-Jui Hsieh. Provable, scalable and automatic perturbation analysis
> on general computational graphs, 2020.

---

### Official Review · AnonReviewer3 · 2020-10-31
**Contributions are well presented and demonstrated**

**Rating:** 6
**Confidence:** 3

**Review:**

This paper address the problem of training robust neural networks with non-uniform perturbation budgets on different input pixels. In practice, a perturbation budget generator is introduced to generate the context-aware perturbation budget (i.e. conditioned on the input) for each pixel of the input image. A “robustness volume” constraint on generated perturbation budgets to control the robustness intensity is also proposed. Extensive experiments on MNIST and CIFAR10 demonstrate the proposed outperform SOTA method under various uniform perturbation budgets.


From my perspective, the writing of this paper is good, and contributions are well presented and demonstrated by extensive experiments. So I vote for accepting

Comments:

- How to determine the hyper-parameters \bar{alpha} and \underline{alpha} for each benchmark is still unknown. Are the final results sensitive to these hyper-parameters? Does it take a high cost to adjust these hyper-parameters for different benchmarks?
- How about the performance by using IBP?

- In Eqn.2, The index i indicates the i-th pixel. But in Eqn.3, it denotes the i-th category label. Please modify this to avoid misunderstanding.

- Since I'm not very well versed with the current baseline and state-of-art for variable robust training of DNN, it would be good to compare with other SOTA methods.

---

> ### Author Response · Authors · 2020-11-22
> **Reply to comments by AnonReviewer3**
>
>
> Dear AnonReviewer3,
>
> We thank the reviewer for the encouraging comment and giving us valuable advice. We address your concerns as follows:
>
> 1. Hyperparameters
>
> Hyperparameters $\underline{\alpha}$ and $\bar{\alpha}$ are to ensure a minimum robustness on each pixel and also prevent the model from over-invariant on some pixels. In our experiments, we intuitively set these hyperparameters and just used some simple values -- 2.0 for $\bar{\alpha}$, 0.5 and 0.125 for $\underline{\alpha}$ on MNIST and CIFAR-10 respectively. We set a smaller $\underline{\alpha}$ on CIFAR-10 because prior works usually use much smaller perturbation radius on this dataset, e.g., 2/255 or 8/255, while we are trying with target robustness volume 32/255. We expect the minimum allowed perturbation budget is smaller than the largest radii used in prior works [1]. If these hyperparameters are changed, models tend to achieve lower verified error if the range constraint defined by $\underline{\alpha}$ and $\bar{\alpha}$ is looser and vice versa. In practice, such hyperparameters can be set according to the need, just as the robustness volume or perturbation radius.
>
> 2. IBP results
>
> As shown in [1], IBP performs worse than CROWN-IBP in all settings, so we propose our framework based on CROWN-IBP. We have also conducted an additional experiment on MNIST with volume $\epsilon_0=0.8$, where verified error by IBP is 27.09 while CROWN-IBP produces a lower (better) verified error of 26.37.
>
> 3. Thank you for pointing out the notation problem. We have modified the notation in the current version of our paper according to your suggestion.
>
> 4. In our paper, we focus on certified defense where the robustness can have a provable guarantee. On certified defense, CROWN-IBP [1] in ICLR 2020 is the state-of-the-art method. Our method has been extensively compared to the models obtained by CROWN-IBP which originally used uniform budgets in experiments.
>
> We thank you again for all the helpful comments by the reviewer, and please kindly let us know if you have any additional comments.
>
> References:
>
> [1] Huan Zhang, Hongge Chen, Chaowei Xiao, Bo Li, Duane Boning, and Cho-Jui Hsieh. Towards stable and efficient training of verifiably robust neural networks. In International Conference on Learning Representations, 2020.

---

### Official Review · AnonReviewer5 · 2020-11-06
**Uses a deep network to learn non-uniform radii for certified defenses, comparisons are not great**

**Rating:** 5
**Confidence:** 5

**Review:**

The paper proposes using a budget generator to generate non-uniform radii for certification. The budget generator is trained jointly with the certified defense to change the shape of the perturbation set while maintaining the volume to improve certified error.

On motivation: the paper could use some more justification or motivation for why we would want to change our perturbation radius during training to maximize certified performance. Typically this the other way around: we have a set we want to defend against, and so the certified defense optimizes for this specified set since we're trying to defend against a particular attack. The setup here is strange in this regard, because rather than adapting a defense to a threat model, the threat model is being adapted to the defense, where the defense is defined by a fixed volume but is otherwise whatever the defender trains it to be. Since an attacker would not conveniently restrict themselves to  the radius learned during training, this doesn't really make much sense from the point of view of certifying robustness to adversarial examples (since it doesn't defend against *all* perturbation sets with the specified volume, it only defends against one which isn't specified a priori).

On comparisons: The authors compare their certified defense with non-uniform budgets to certified defenses with uniform budgets. In its current form, this is completely incomparable: the uniform budgets are trained and certified with uniform radius, while the learned budgets are trained and certified with learned budgets. Since the learned budgets are almost certainly different from the uniform budgets, these are completely different threat models. It would be much more informative to the reader to report results on *both* types of budgets for *both* models instead, rather than only showing half of the story. Specifically, this means
(a) evaluating all the models trained with learned budgets using the uniform budgets that are more typical in the literature and reflect what they were actually trained for (e.g. <= 0.4 for MNIST, <= 8/255 for CIFAR10)
(b) evaluating the models trained with uniform budgets using the learned budgets  to compare against the models trained with learned budgets
It's quite possible that the learned budgets, while capable of certifying more volume due to the changed threat model, comes at the cost of worse certified performance for a uniform bound of a *smaller* volume (e.g. at uniform radius 0.1 or 0.3 for MNIST, as is commonly reported in the literature). This would also help provide a more realistic and fair comparison to certified defenses with uniform budgets: the current tables report certified accuracy at an extremely large radii well beyond what they were trained for, and so this winds up being a rather a misleading comparison that is not very useful.

On related work and comparisons thereof: The authors seem to be unaware of the ICML 2019 publication "On Certifying Non-Uniform Bounds against Adversarial Attacks" by Liu et al., which has studied the problem of certifying non-uniform bounds which maximize the volume (exactly the same type of bound studied in this work). There is still a difference, in that they optimize the budget to maximize the volume rather than use a generator to produce perturbation budgets, and do not train. Nonetheless, this is arguably the most relevant work and has been out for quite a while, and so it would be fair to expect some sort of comparison here. For example, a reasonable experiment could be to simply calculate the non-uniform bounds from Liu et al. on the model trained with a non-uniform budget vs the uniform budget.

Minor comment: The authors mention that the joint training of the classifier and the perturbation budget generator is somehow more difficult for PGD adverarial training "as it is not fully differentiable w.r.t. perturbation budgets". I don't quite get what the authors are trying to say here. My understanding is that the authors perform joint training by backpropagating the robust loss through both the classifier and the perturbation budget generator, since there is no auxiliary loss for the perturbation budget generator. Shouldn't this imply that the standard loss is in fact differentiable with respect to the perturbation budgets, and so PGD is just as applicable as before?

Update:
I thank the authors for their response. I've read the other reviews as well, and indeed R2 had similar concerns to my own. I'm glad to see the more comprehensive comparison to Liu et al., which paints a fuller picture of the effects and trade-offs of the approach.

The argument behind the motivation, however, feels much like setting up a straw man for Lp robustness. For example, the authors argue that their approach is label and semantics preserving unlike uniform perturbations; however this is quite frankly only the case for extremely large perturbations in MNIST-like settings which are unrealistic by design (most papers do not consider such large radii for exactly this reason). Uniform perturbations seen commonly in CIFAR10/Imagenet settings are practically invisible and consequently are equally semantics preserving and close to human perception. If the authors do wish to pursue this argument that these are truly more semantics preserving, then this needs to be backed by evidence. The authors weakly suggest this is the case because the budgets look similar to the content in the images. However, this does not imply that an adversarial attack within this budget is label preserving (i.e. many of the presented examples have large budgets in the background directly adjacent to the label-content of the image, which can easily change how the content looks), and so this needs to be justified carefully if this claim is to be made.

The authors also incorrectly equate the restrictions imposed on an attacker from learned perturbation radii to that of a uniform radius. These are *not* equivalent, especially in the security setting where these are night and day; the first amounts to the defender choosing the rules of the game that work optimally for them, whereas the latter is a *defense agnostic* rule that both the defender and attacker must obey. This is a significantly easier setting for the defender that needs to be properly motivated, as restricting an adversary to a fixed perturbation set is inherently different from restricting the adversary to a fixed perturbation set that the defender gets to choose. The reason why one would want to maximize certification volume needs to be properly motivated, as it is no longer applicable to the usual adversarial security setting and comes at a cost to the usual robustness considerations.

To recognize the addition of the necessary comparison to past work, I have improved my score slightly. However, I would still argue that this is below the threshold, as their central claim of learning *semantic preserving* perturbation budgets is not justified despite being a central component of the paper, as well as the motivation for why it's considered beneficial to choose the most easily certified volumes for robustness in the first place (and certainly not helpful from a security perspective).

---

> ### Author Response · Authors · 2020-11-22
> **Differentiability for PGD training (4/4)**
>
> ### Differentiability for PGD training
>
> The loss is technically differentiable to the perturbation budgets however it is much more complicated. For PGD training, for example, if we run $N$ steps, each step we add a noise roughly $\frac{\epsilon}{N}$, so the $\epsilon$ is used by every step of PGD. Eventually, to get the correct gradient w.r.t $\epsilon$, we will need to backprop through the $N$ PGD update steps since $\epsilon$ affects every step of PGD. In contrast, in certified defense we use Eq. (7) where loss is a direct function of $\epsilon$. In fact, we did some initial experiments with adversarial training and found that it is possible but more tricky to train the perturbation generator. Nevertheless, the main goal of our paper is to present the idea of learning a non-uniform and context dependent perturbation budget, and we use certified defense to demonstrate it just for technical convenience.
>
> ### Conclusion
>
> We hope the reviewer can understand the motivation behind our work better now and your concerns are addressed. We would love to answer any additional questions you may have.
>
> References:
>
> [1] Tramèr F, Behrmann J, Carlini N, et al. “Fundamental tradeoffs between invariance and sensitivity to adversarial perturbations” ICML 2020
>
> [2]Liu, Chen, Ryota Tomioka, and Volkan Cevher. “On certifying non-uniform bound against adversarial attacks.” ICML 2019.
>
> [3] Yang, Jiancheng, Shi, Rui, and Ni, Bingbing. Medmnist classification decathlon: A lightweight automl benchmark for medical image analysis. arXiv preprint arXiv:2010.14925
> , 2020.
>
> [4] Huan Zhang, Hongge Chen, Chaowei Xiao, Bo Li, Duane Boning, and Cho-Jui Hsieh. Towards stable and efficient training of verifiably robust neural networks. In International Conference on Learning Representations, 2020.

---

> ### Author Response · Authors · 2020-11-22
> **New experiments and comparisons (3/4)**
>
> ### New experiments and comparisons
>
> We provide additional experiments to show why the contextual perturbation budget is necessary and demonstrate that our method has superior performance over existing methods.
>
> 1. Evaluating models using Liu et al, 2019’s method
>
> We have reproduced the algorithm proposed in Liu et al., 2019 with their open-source code, and the new results have been updated to **Table 1 and Table 2**. On MNIST, when models are evaluated by Liu et al., the model trained with $\epsilon=0.4$ uniform budget has a little lower verified error (5.88 v.s. 7.97) than the model with learned budgets. However, as the target volume is increased, on $\epsilon_0=0.6$ and $\epsilon_0=0.8$, models trained with uniform budgets totally fail with verified error 100.0%, while our models can still achieve reasonable verified errors (13.93 on $\epsilon_0=0.6$ and 25.77 on $\epsilon_0=0.8$). On CIFAR-10, our models can **achieve lower verified errors than models trained with uniform budgets**, even if the uniform budget models are evaluated with Liu et al.
>
> 2. Evaluating models on uniform budgets
>
> As requested by the reviewer, we have updated our Table 1 and Table 2. Now for models trained with both learned budgets and uniform budgets, we evaluate them under three methods: fixed uniform budgets, learned budgets, and non-uniform certification by Liu et al.
>
> It is true that models with learned budgets tend to have higher verified errors when they are evaluated with uniform budgets. However, achieving good robustness on uniform budgets is not the goal of the paper, nor it is the ultimate goal of robust machine learning. We have argued in the “Motivations and Justifications” section above, and we believe that using non-uniform and contextual budgets is a better setting than uniform budgets. For example in the MNIST $\epsilon=0.4$ setting, [1] showed that we can actually find many images within this $\ell_\infty$ perturbation budget that *changed the ground-truth label*.
>
> Nevertheless, if we want to maintain a satisfactory verified accuracy on a smaller uniform budget and also pursue a robustness under learned non-uniform budgets, we may set $\underline{\alpha}$ which controls the minimum allowed budget for each pixel. To show an example, we train a model on MNIST with $\epsilon_0=0.5$ and $\underline{\alpha}=0.6$, which means the minimum budget of each pixel is 0.3. As a result, verified error on non-uniform budgets is 10.46, while the verified error on uniform $\epsilon=0.3$ budget is 9.96, which is close to the 9.40 verified error reported in CROWN-IBP[4] on the same “DM-small” model trained with uniform budgets. And if $\epsilon_0=0.6, \underline{\alpha}=0.5$, the verified error on uniform $\epsilon=0.3$ is 11.22 which is also not far away from 9.40, while now our model has a robustness on a volume as large as $0.6^n (n=28*28)$.
>
> 3. MedMNIST
>
> MedMNIST [3] is a recently developed medical imaging dataset. There are 10 sub-datasets and we adopt the OrganMNIST(axial) subset consisting of body organ CT images (please refer to Appendix B for more details). On target robustness volume $\epsilon_0=16/255$, we train a model with learned perturbation budgets and uniform budgets respectively. For the model with learned budgets, the clean error is 21.8 and the verified error is 50.6, while for the model with uniform budgets, the clean error is 28.0 and the verified error is 57.5. Using learned contextual perturbation budgets, we are able to obtain a model with lower clean and verified errors under the same target robustness volume. We also visualize learned budgets in Figure 6 in Appendix B.

---

> ### Author Response · Authors · 2020-11-22
> **Comparisons to the work of Liu et al. (2/4)**
>
> ### Comparisons to the work of Liu et al.
>
> We apologize for not citing this insightful paper. We have updated our paper which includes discussions on Liu et al [2]. We have made significantly different contributions compared to Liu et al. as the reviewer mentioned, “they optimize the budget to maximize the volume rather than use a generator to produce perturbation budgets, and do not train”. We also provide additional experiments to demonstrate that our method is capable of training models with larger robustness volumes and outperforming models trained with uniform budgets, even if the models trained with uniform budgets are evaluated with Liu’s method. We summarize our main differences below:
>
> 1. We are a robust training method:
>
> In Liu’s work, they propose to compute a largest possible certified volume for a **pretrained model**. They don’t update model parameters. In contrast, our method can **train the classifier jointly with the generator**, so that our model can learn semantically meaningful perturbation budgets.
>
> Additionally, Liu’s method cannot be extended to an efficient training method, as their method requires solving a constrained optimization problem via the Lagrangian method for each input example using a large number of iterations, which is too costly for training. We introduce a perturbation generator network with nearly no additional cost in training compared with existing uniform certified defense methods.
>
> 2. Different problem formulation:
>
> Liu’s method **maximizes the volume** of the perturbation budget for a fixed network under the constraint that the network prediction is correct within this perturbation region, while our method fixes the volume of the perturbation budget and jointly trains a perturbation generator with the classifier to **maximize the robust accuracy**, which matches the training objective of prior works for certified defense.
>
> 3. Certification approach and efficiency:
>
> As for certification in the inference stage, Liu’ method is still too inefficient as a large number of optimization iterations are required for every batch, while we only need a forward propagation with a small perturbation generation network, and use the efficient (CROWN-)IBP method to obtain certificates.
>
> 4. Semantic and contextual perturbation:
>
> Our visualization of the generated perturbation budget and experiments on Watermarked MNIST and Doubled Cifar-10 (see section 4.2,4.3 for details) demonstrates that our perturbation budgets indeed learns contextual knowledge. This implies that training the classifier and the generator jointly enables the budget generator to capture the contextual information of input images, rather than just optimizing training objectives, while Liu’s work did not have such analysis.
>
> 5. Additional experiments:
>
> In Section 4, we conduct additional experiments to evaluate models with the method in Liu et al., as also shown in the “New experiments and comparisons” section in this response. Our method is capable of training robust models on larger robustness volumes and achieving lower verified errors.

---

> ### Author Response · Authors · 2020-11-22
> **Reply to AnonReviewer5: Motivations and Justifications (1/4)**
>
> **(There are currently 4 posts in total for our author response to AnonReviewer5. This is the first one, and the other 3 are presented below this one.)**
>
> Dear AnonReviewer5,
>
> Thank you for thoroughly reading our paper and offering us insightful suggestions. In our response, we provide more motivations and justifications, provide comparison experiments as you requested (including comparisons to Liu et al.), and include some initial results on MedMNIST (a new medical imaging dataset) where the non-uniform perturbation is important.
>
> Here are our detailed responses to your questions and concerns:
>
> ### Motivations and Justifications
>
> The reviewer’s main concern is about the threat model. As mentioned by the reviewer, “an attacker would not conveniently restrict themselves to the radius learned during training”. However, in fact, a real attacker would also not restrict them to the common $\ell_p$ norm radius. The use of $\ell_p$ norm is mostly for mathematical convenience, and is largely inappropriate for many realistic scenarios.
>
> First, the perturbations from real world attackers have to **follow the semantics of the image** to generate meaningful attacks. The goal of our perturbation generator is to learn “what is the maximal allowable perturbation **that preserves ground-truth label given the current context**”? An attacker is successful only if it can fool the network while preserving the ground truth label of the image at the same time. If the attacker completely disregards the context of the input, the ground truth label of the perturbed image can differ from that of the original image (e.g., see the “invariance adversarial examples” in [1]), and this is not an adversarial attack anymore. By comparison, our contextual perturbation budget gives detailed characterization for the ability of the attacker by generating perturbation radius for each pixel. Ideally, if a model is robust within our generated perturbation budget, we can expect that an attacker can hardly change the model output and preserve the ground-truth label simultaneously. This is in fact a more realistic and useful threat model than $\ell_p$ norm.
>
> Another important motivation for our work is that real datasets have varied sensitivity on each feature, and our method enables robust training on these datasets effortlessly. In contrast, the $\ell_p$ threat model is problematic because it is hard to define an uniformly good perturbation budget, and such a budget can be feature dependent and contextual so not easily defined manually. This prevents the usage of robust training in practical settings. For example, in a medical imaging setup, the difference between health and unhealthy tissues may be subtle and can only be recognized by experts. Using an uniform $\ell_\infty$ perturbation for robust training is likely to destroy the subtle but important features, reducing clean accuracy. Training on a uniform radius can harm the clean accuracy of the model. Empirically, we have reported initial results on the MedMNIST dataset to support this point (see details below).
>
> Ultimately, **the goal of robust machine learning is to learn a classifier close to human perception, rather than “robust” to certain $\ell_p$ norms**. Human perception is non-uniform (humans focus on important features even though these features can be sensitive to small noise) and context dependent (what part of image is important heavily depends on what’s on the image). Admittedly, the perturbation generation process in our paper still has large room to improve to eventually match human perception. However, we believe it is import to think beyond the $\ell_p$ robustness model for robust training, especially in our work we consider *context dependent* perturbation budget, which was never proposed by prior works. We hope our paper can help the community understand the limitation of $\ell_p$ norm and think beyond it.

---

### Decision · Program_Chairs · 2021-01-07
**Final Decision**

**Decision:**

Reject

**Comment:**

Reviewers raised various concerns about the motivation, unclear justification of the idea and claim, insufficient comparison with related work, and weak experimental results. While authors had made efforts to improve some of these issues in the rebuttal, the revision was not satisfied for publication quality. Overall, the paper has some interesting idea, but is not ready for publication.